# The diagnostic performance of CA125 for the detection of ovarian and non-ovarian cancer in primary care: A population-based cohort study

**Garth Funston**[1]*, **Willie Hamilton**[2], **Gary Abel**[2], **Emma J. Crosbie**[3,4], **Brian Rous**[5], **Fiona M. Walter**[1]

**1** The Primary Care Unit, Department of Public Health and Primary Care, University of Cambridge, Cambridge, United Kingdom, **2** University of Exeter, Exeter, United Kingdom, **3** Gynaecological Oncology Research Group, Division of Cancer Sciences, University of Manchester, Manchester, United Kingdom, **4** Department of Obstetrics and Gynaecology, Manchester University NHS Foundation Trust, Manchester Academic Health Sciences Centre, Manchester, United Kingdom, **5** National Cancer Registration and Analysis Service, Public Health England, Cambridge, United Kingdom

* gf272@medschl.cam.ac.uk

**Data Availability Statement:** Data cannot be shared publicly as they are analysed under licence. All data used in this study are available from CPRD

## Abstract

### Background

The serum biomarker cancer antigen 125 (CA125) is widely used as an investigation for possible ovarian cancer in symptomatic women presenting to primary care. However, its diagnostic performance in this setting is unknown. We evaluated the performance of CA125 in primary care for the detection of ovarian and non-ovarian cancers.

### Methods and findings

We studied women in the United Kingdom Clinical Practice Research Datalink with a CA125 test performed between 1 May 2011–31 December 2014. Ovarian and non-ovarian cancers diagnosed in the year following CA125 testing were identified from the cancer registry. Women were categorized by age: <50 years and ≥50 years. Conventional measures of test diagnostic accuracy, including sensitivity, specificity, and positive predictive value, were calculated for the standard CA125 cut-off (≥35 U/ml). The probability of a woman having cancer at each CA125 level between 1–1,000 U/ml was estimated using logistic regression. Cancer probability was also estimated on the basis of CA125 level and age in years using logistic regression. We identified CA125 levels equating to a 3% estimated cancer probability: the "risk threshold" at which the UK National Institute for Health and Care Excellence advocates urgent specialist cancer investigation.

A total of 50,780 women underwent CA125 testing; 456 (0.9%) were diagnosed with ovarian cancer and 1,321 (2.6%) with non-ovarian cancer. Of women with a CA125 level ≥35 U/ml, 3.4% aged <50 years and 15.2% aged ≥50 years had ovarian cancer. Of women with a CA125 level ≥35 U/ml who were aged ≥50 years and who did not have ovarian cancer, 20.4% were diagnosed with a non-ovarian cancer. A CA125 value of 53 U/ml equated to a 3% probability of ovarian cancer overall. This varied by age, with a value of 104 U/ml in

(www.cprd.com). Permission to access data is through the Independent Scientific Advisory Committee (ISAC, contact via: isac@cprd.com). Derived data used to prepare article figures are available from the University of Cambridge Repository: www.repository.cam.ac.uk (https://doi.org/10.17863/CAM.56363).

**Funding:** This research arises from the CanTest Collaborative, which is funded by Cancer Research UK [C8640/A23385], of which GF is Clinical Research Fellow, GA is the Senior Statistician, and WH and FMW are Directors. The study was also funded by the National Institute of Health Research (NIHR) School of Primary Care Research [FR17 424] (GF, FMW). EJC is supported through the NIHR Manchester Biomedical Research Centre [IS-BRC-1215-20007]. The funders of this study had no role in the study design, data collection, data analysis, data interpretation, or writing of the report.

**Competing interests:** The authors have declared that no competing interests exist.

**Abbreviations:** AUC, area under the curve; CA125, cancer antigen 125; CI, confidence interval; CT, computed tomography; GP, general practitioner; IT, information technology; NICE, National Institute for Health and Care Excellence; NPV, negative predictive value; PPV, positive predictive value; ROCkeTS, Refining Ovarian Cancer Test Accuracy Scores.

40-year-old women and 32 U/ml in 70-year-old women equating to a 3% probability. The main limitations of our study were that we were unable to determine why CA125 tests were performed and that our findings are based solely on UK primary care data, so caution is need in extrapolating them to other healthcare settings.

## Conclusions

CA125 is a useful test for ovarian cancer detection in primary care, particularly in women ≥50 years old. Clinicians should also consider non-ovarian cancers in women with high CA125 levels, especially if ovarian cancer has been excluded, in order to prevent diagnostic delay. Our results enable clinicians and patients to determine the estimated probability of ovarian cancer and all cancers at any CA125 level and age, which can be used to guide individual decisions on the need for further investigation or referral.

## Author summary

### Why was this study done?

- CA125 is widely used as a test for ovarian cancer in women presenting with relevant symptoms, both in the UK and internationally.
- CA125 has been extensively evaluated in the specialist care setting and in screening studies but little was known about its diagnostic performance in primary care, where most patients with ovarian cancer first present.

### What did the researchers do and find?

- We evaluated the diagnostic performance of CA125 in 50,780 women undergoing testing in English general practice.
- Of women with CA125 levels above the current abnormal cut-off, 10.1% were diagnosed with ovarian cancer and a further 12.3% with another form of cancer.
- Cancer was more common in women with abnormal CA125 levels if they were ≥50 years of age.
- We developed models to estimate the probability of ovarian cancer and all cancer based on a woman's age and CA125 level.

### What do these findings mean?

- Clinicians should consider the possibility of non-ovarian cancer, in addition to ovarian cancer, in women with high CA125 levels.
- Our models will enable patients and clinicians to determine the estimated probability of cancer based on an individual's age and CA125 level.
- This information can be used to help make decisions about the need for further investigation or urgent referral to a specialist.

- The findings should also help inform guidelines, as they will allow recommendations for further testing to be made on the basis cancer probability rather than a single CA125 cutoff.

## Introduction

Ovarian cancer is the eighth most common cancer to affect women worldwide, accounting for over 384,000 deaths in 2018 [1]. Survival depends on stage at diagnosis, with five-year net survivals of 93% for stage I, 68% for stage II, 27% for stage III, and 13.4% for stage IV disease [2]. Most women are diagnosed following a symptomatic presentation [3], and, in healthcare systems in which general practitioners (GPs) play a gatekeeping role, this initial presentation usually takes place in primary care [4].

Symptoms can occur at all stages of ovarian cancer [5]. However, they are usually nonspecific and are common in women without ovarian cancer, so they only have modest positive predictive values for the disease [5,6]. The serum biomarker cancer antigen 125 (CA125) is widely used in countries around the world, including the United States, Australia, Canada, and Ireland, as an investigation for ovarian cancer in symptomatic women presenting to primary care [7]. In 2011, the United Kingdom National Institute for Health and Care Excellence (NICE) recommended that women with symptoms of possible ovarian cancer be tested for CA125 in primary care, with further investigation advocated in those with CA125 levels ≥35 U/ml [8]. The chosen cutoff of 35 U/ml is the conventional upper limit of normal for CA125 and derives from a study in which 1% of healthy women and 82% of patients with ovarian cancer had a CA125 level >35 U/ml [9].

CA125 has been studied extensively in screening studies and in women in secondary care with pelvic masses but not in women presenting with symptoms of possible ovarian cancer in primary care. The NICE recommendations on CA125 testing for symptomatic women are based on extrapolated secondary care and screening data rather than primary care data [8]. The performance characteristics of a test vary with disease prevalence, disease severity, and the prevalence of other conditions that elevate test levels, so it is important to evaluate CA125 within the intended population [10].

When evaluating the diagnostic performance of a test such as CA125, it is standard practice to report accuracy characteristics, including the positive predictive value (PPV), after applying a particular cutoff. However, the PPV provides the "average" probability of disease for all women with a test level at or above the set cutoff rather than the probability of disease at a given test level. Knowledge of the probability of cancer at any given CA125 level is likely to be more clinically useful than the PPV, as it would allow patients and clinicians to interpret their individual CA125 test results, which could help guide decisions on the need for further investigations. NICE revised their cancer guidance in 2015, using a "risk threshold" of ≥3% as the threshold for urgent cancer investigation in symptomatic women, but ovarian cancer guidance, including the chosen CA125 cut-off of 35 U/ml, remained unchanged [11]. Knowledge of the estimated probability of cancer at each CA125 level could help inform health policy both in the UK and internationally.

The primary aim of this study was to explore the relationship between CA125 level and ovarian cancer probability, to identify the CA125 level at which a 3% probability of ovarian cancer was reached. Given the nonspecific nature of ovarian cancer symptoms, and reports

indicating CA125 is commonly elevated in other cancers [12,13], a second aim was to explore the relationship between CA125 level and the probability of all cancers. To allow comparison with existing literature on CA125 diagnostic accuracy, we also calculated conventional test performance characteristics, including PPV, sensitivity, and specificity, applying the standard cutoff ($\geq$35U/ml).

## Methods

### Ethics statement

The study was approved by the Independent Scientific Advisory Committee (ISAC) for the Medicines and Healthcare Products Regulatory Agency (protocol number 18_184). All data were provided to researchers in an anonymized form, and individual consent was not required.

### Data source

This was a retrospective cohort study using linked data from the Clinical Practice Research Datalink (CPRD) GOLD dataset and the National Cancer Registration and Analysis Service (NCRAS). The CPRD GOLD dataset contains anonymized, coded, primary care data including demographics, laboratory results, symptoms, and diagnoses for around 11 million patients. It is broadly representative of the UK population [14]. The NCRAS (English cancer registry) collects cancer registration data on patients, including detailed information on tumor topography, stage, and date of diagnosis. NCRAS obtains data from multiple sources including hospitals, GP surgeries, and death certificates and reports a near 100% case ascertainment [15]. Linkage of CPRD and NCRAS data was performed at a patient level by a third party, National Health Service (NHS) Digital [16]. As NCRAS only collects details of cancers diagnosed in England, the study was restricted to English general practices. The approved ISAC protocol, which covers several linked studies, is included in the S1 Text and S2 Text.

This report conforms to the STARD and RECORD statements [17,18]. A completed STARD checklist is included with this article (S3 Text).

### Participants

We included women with a code for CA125 measurement in primary care (S1 Table) between 1 May 2011 and the 31 December 2014. There has never been a national ovarian cancer screening program in the UK, and the only indication for CA125 testing in English primary care is a presentation with a symptom of possible ovarian cancer. As such, we assumed that CA125-tested women were symptomatic.

Women who were <18 years old or registered at a GP practice not deemed "up-to-standard" on data quality by CPRD on the date of their first CA125 test during this period were excluded [14]. Women with a record of ovarian cancer in NCRAS data on or before the CA125 test date were also excluded, as were women with a CA125 test in the 12 months before the first CA125 test during the study period. Only CA125 entries recorded in standard equivalent units of CA125 measurement (U/ml, IU/ml, KU/L, KIU/L) were accepted. Although NICE recommends a CA125 cutoff of $\geq$35 U/ml, individual laboratory cutoffs varied. We excluded CA125 values associated with spurious laboratory cutoffs (245, 420, and 455 U/ml) and those where no cutoff was given. Subsequent sensitivity analyses, including CA125 entries recorded in all units and associated with all laboratory upper cutoffs, had minimal impact on our results. The first CA125 test during the study period was used in analyses.

## Clinical outcomes

**Primary outcome.** Our primary clinical outcome was the diagnosis of ovarian cancer, as recorded using International Classification of Diseases (ICD)-10 codes in NCRAS data, in the 12 months following the initial CA125 test. With reference to the International Federation of Gynaecology and Obstetrics (FIGO) and WHO classifications [19,20], we defined ovarian cancer as an ovarian malignancy (C56), a fallopian tube malignancy (C57.0), a peritoneal malignancy (C48.1, C48.2), or a neoplasm of uncertain behavior of the ovary (D39.1). We assumed that cancer diagnosed within 12 months of the initial CA125 test was present at the time of testing. It is possible that incidental ovarian cancers may arise and be diagnosed in the year following testing or that it may take longer than 1 year from presentation in primary care to diagnosis. A period of 1 year, which has been used widely in similar studies [21,22], was chosen as a compromise between minimizing the inclusion of incidental cancers and maximizing the inclusion of relevant cancers.

Our primary outcome included borderline ovarian tumors, as these are treated collectively with invasive tumors in NICE recommendations on CA125 testing and generally require surgical management [23]. Although their timely detection in symptomatic women is important, borderline tumors are less likely to cause an elevation in serum CA125, and their prognosis is very good even if detected late [23]. We therefore performed a subanalysis in which invasive ovarian cancer formed the outcome.

**Secondary outcome.** Our secondary outcome was the diagnosis of non-ovarian cancers. The earliest record of cancer, excluding nonmelanoma skin cancers, was identified in the 12 months following initial CA125 testing in women without ovarian cancer. We refer to this group of cancers as "non-ovarian cancer." Where we discuss the combined non-ovarian and ovarian cancer groups, we use the term "all cancers."

**Descriptive outcomes.** In order to report the symptoms that may have triggered CA125 testing, symptoms coded in the 30 days before CA125 testing were identified from CPRD data using a code list of ovarian cancer symptoms from current NICE guidelines [11].

Ovarian cancer stage was determined using the Tumor Nodes Metastasis (TNM) staging system or, where not recorded, the FIGO staging system, and the proportions of ovarian cancers at each stage identified [19].

The histology of invasive ovarian tumors was identified from NCRAS data and categorized on the basis of ICD10 codes.

## Statistical analysis

We calculated the PPV, negative predictive value (NPV), sensitivity, and specificity of CA125 for ovarian cancer at or above the current cut-off (35 U/ml). A nonparametric receiver operator characteristic (ROC) curve was constructed, and the area under the curve (AUC) determined. This analysis was repeated for invasive ovarian cancer and all cancers combined. After excluding ovarian cancer patients, it was also repeated for non-ovarian cancer. As ovarian cancer incidence is greater in older women and most cases occur in women post-menopause, we repeated all analyses for women <50 years and ≥50 years of age [24].

We used logistic regression to examine the relationship between CA125, as a continuous variable, and ovarian cancer diagnosis. CA125 level was highly skewed, and so it was log-transformed prior to regression analysis. Log CA125 was centered on a value of 3, the closest integer to the mean. The relationship between log CA125 level and ovarian cancer was nonlinear. To account for this, we used restricted cubic splines. As recommended by Harrell [25], we compared the Akaike Information Criterion (AIC) for models containing 3, 4, and 5 knots. The 5-knot model produced the smallest AIC and so was taken forward. Knots were placed at

standard, equally spaced percentiles of the marginal distribution of the variable (S4 Text) [25]. This regression model was used to predict the odds of cancer for a range of CA125 levels (1–1,000 U/ml), which were then converted into probabilities.

The logistic regression analysis was repeated for the <50 years and ≥50 years age groups. Significant differences were noted between these groups in terms of the estimated ovarian cancer probabilities. Given this, and on the recommendation of a peer reviewer, we constructed a multivariable regression model including age in years (mean centered) as a continuous variable and CA125 level, applying the same approach as described here previously. Five knots were included for each variable (S4 Text). This regression model was used to predict the odds of ovarian cancer for CA125 levels (1–1,000 U/ml) in women of different ages. Results for women aged 30, 40, 50, 60, 70, and 80 years of age are presented as examples in this paper.

All the aforementioned steps were repeated for our secondary outcome and for the invasive cancer subanalysis. Full details of all models are included in S4 Text.

**Statistical software.** All analyses were performed in Stata version 15.1 (StataCorp, www.stata.com). The DIAGT module was used to calculate summary diagnostic accuracy statistics [26]. All confidence intervals (CI) are reported at the 95% level.

## Results

After exclusions, our cohort consisted of 50,780 women (**Fig 1**).

The ovarian cancer incidence in the cohort was 0.9% and was 3 times higher in the ≥50 years group than the <50 years group (**Table 1**). The median interval between CA125 testing and ovarian cancer diagnosis was 42 days (interquartile range: 25–62 days) and the mean patient age was 56 years (range: 18–102 years).

### Cancer stage

Of the 456 ovarian cancers, 172 (37.7%) were stage I or II, and 209 (45.8%) were stage III or IV. No stage was recorded in 75 (16.4%) cases (S2 Table).

### Cancer morphology and histology

Of the ovarian cancers diagnosed, 21.5% ($n$ = 98) were borderline tumors. The proportion of malignancies that were borderline varied with age, with 50% of tumors in the <50 years group and 15.4% in the ≥50 years group being borderline (S3 Table). Serous epithelial tumors were the most common tumor type, accounting for 48.6% ($n$ = 174) of invasive tumors. In the <50 years group, 12.5% ($n$ = 5) of invasive tumors were of nonepithelial origin compared with 2.5% ($n$ = 8) in the ≥50 years group.

### Recorded symptoms

Symptoms of possible ovarian cancer were recorded for 24,269 women (47.8%) on the same day or in the 30 days preceding CA125 testing; the most common was abdominal pain (**Table 2**). Multiple symptoms were recorded in 1,477 (6.1%) women.

### Diagnostic performance applying the standard cutoff (≥35 U/ml)

The diagnostic performance characteristics of CA125 were calculated after applying the standard cutoff (≥35 U/ml) (**Table 3**). At or above the 35 U/ml cutoff, CA125 demonstrated a PPV of 10.1% (95% CI 9.1–11.2), an NPV of 99.8% (95% CI 99.7–99.8), a sensitivity of 77.0% (95% CI 72.8–80.8) and a specificity of 93.8% (95% CI 93.6–94.0) for ovarian cancer. The AUC was 0.92 (95% CI 0.90–0.93). The AUC was greater in the ≥50 years group (AUC: 0.93,

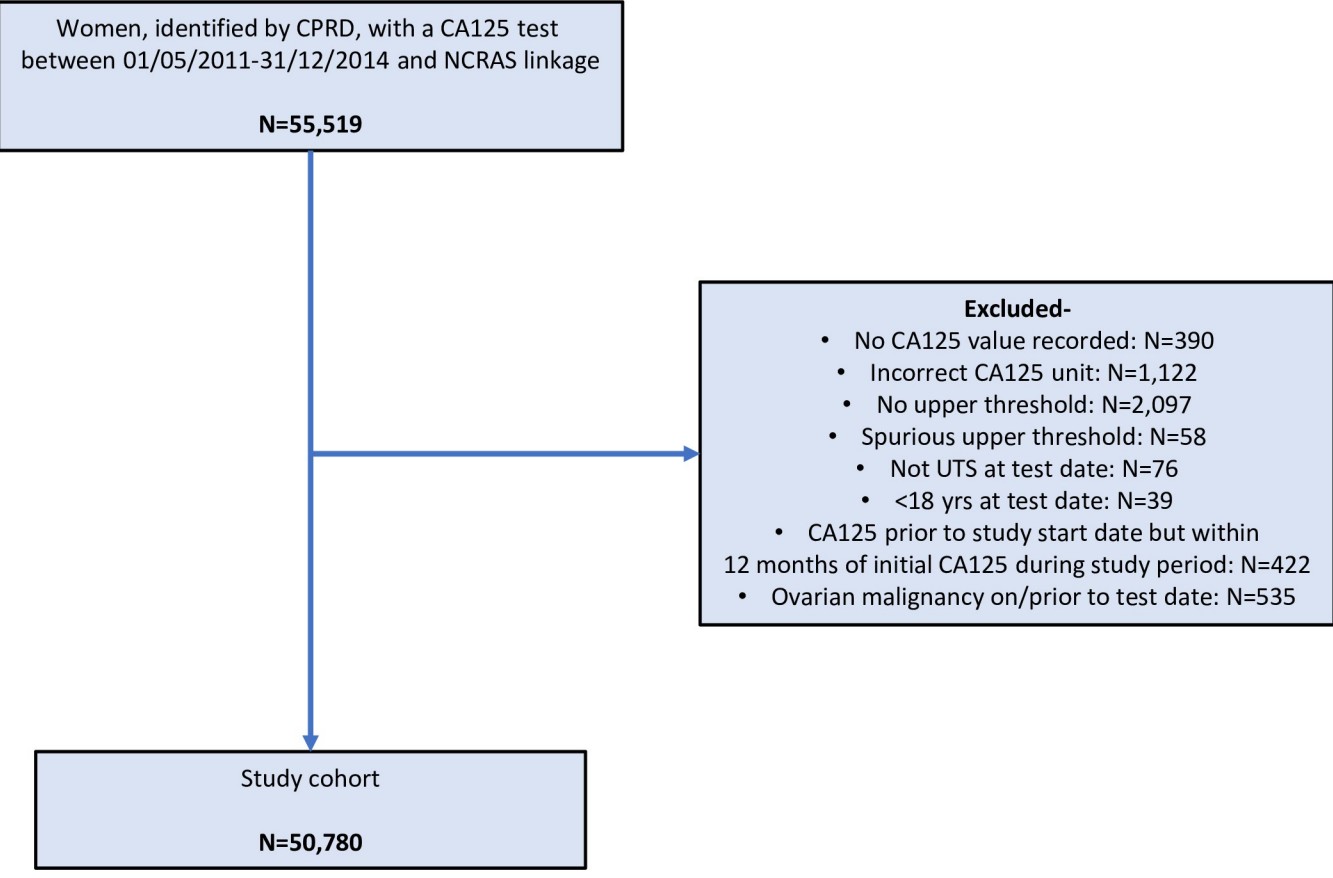

**Fig 1. Flow diagram illustrating the identification of the study cohort and application of exclusion criteria.** UTS is a quality metric, provided by CPRD, which indicates if the data from a GP practice are of sufficient quality to be used in research [14]. CA125, cancer antigen 125; CPRD, Clinical Practice Research Datalink; NCRAS, National Cancer Registration and Analysis Service; UTS, up to standard.

95% CI 0.92–0.95) than the <50 years group (AUC 0.86, 95% CI 0.82–0.91) and the PPV, sensitivity and specificity were also higher in the ≥50 group.

When the outcome was restricted to invasive ovarian cancers, CA125 demonstrated a slightly lower PPV (8.8%, 95% CI 7.8–9.8), and a higher sensitivity (84.9%, 95% CI 80.8–88.5) (**Table 3**).

Of the 50,324 women without ovarian cancer, 1,321 (2.6%) were diagnosed with a non-ovarian cancer. The incidence of non-ovarian cancers in women with a CA125 <35 U/ml was

**Table 1. Patient numbers, incidence of raised CA125 tests (≥35 U/ml) and cancer incidence by age group.**

|  | <50 years | ≥50 years | Overall cohort |
|---|---|---|---|
| Number of patients, N | 19,694 | 31,086 | 50,780 |
| Raised (≥35 U/ml) CA125, N (%) | 1,482 (7.5) | 1,986 (6.4) | 3,468 (6.8) |
| Ovarian cancers, N (%) | 80 (0.4) | 376 (1.2) | 456 (0.9) |
| Non-ovarian cancer, N (%) | 161 (0.8) | 1,160 (3.7) | 1,321 (2.6) |

CA125, cancer antigen 125.

**Table 2. Ovarian cancer symptoms and signs coded in the 30 days prior to CA125 testing.**

| Symptom/sign | Patients, N (%) |
|---|---|
| Abdominal pain | 11,933 (49.2) |
| Abdominal distension or bloating | 5,686 (23.4) |
| Change in bowel habit | 2,866 (11.8) |
| Fatigue | 1,692 (7.0) |
| Pelvic pain | 1,632 (6.7) |
| Weight loss | 913 (3.8) |
| Urinary frequency | 552 (2.3) |
| Abdominal or pelvic mass | 419 (1.7) |
| Loss of appetite | 113 (0.5) |
| Urinary urgency | 86 (0.4) |
| Ascites | 26 (0.1) |

% is the proportion of patients with a given symptom out of the total number of patients who have a coded symptom (N = 24,269). Categories are not mutually exclusive: a patient may have had more than 1 symptom coded.

CA125, cancer antigen 125.

**Table 3. Performance characteristics of CA125 for ovarian cancer, invasive ovarian cancer, non-ovarian cancers and all cancers.**

| Cancer | Group | PPV, % (95% CI) | NPV, % (95% CI) | Sensitivity, % (95% CI) | Specificity, % (95% CI) | AUC (95% CI) |
|---|---|---|---|---|---|---|
| Ovarian | All ages | 10.1 (9.1–11.2) | 99.8 (99.7–99.8) | 77.0 (72.8–80.8) | 93.8 (93.6–94.0) | 0.92 (0.90–0.93) |
| | <50 years | 3.4 (2.5–4.4) | 99.8 (99.8–99.9) | 62.5 (51.0–73.1) | 92.7 (92.3–93.1) | 0.86 (0.82–0.91) |
| | ≥50 years | 15.2 (13.6–16.8) | 99.7 (99.7–99.8) | 80.1 (75.7–84.0) | 94.5 (94.3–94.8) | 0.93 (0.92–0.95) |
| Ovarian: invasive | All ages | 8.8 (7.8–9.8) | 99.9 (99.9–99.9) | 84.9 (80.8–88.5) | 93.7 (93.5–93.9) | 0.94 (0.92–0.96) |
| | <50 years | 2.0 (1.3–2.8) | 99.9 (99.9–100) | 72.5 (56.1–85.4) | 92.6 (92.2–93.0) | 0.88 (0.82–0.95) |
| | ≥50 years | 13.8 (12.4–15.4) | 99.9 (99.8–99.9) | 86.5 (82.2–90.0) | 94.4 (94.2–94.7) | 0.95 (0.93–0.97) |
| Non-ovarian | All ages | 12.3 (11.2–13.5) | 98.0 (97.9–98.1) | 29.1 (26.6–31.6) | 94.4 (94.2–94.6) | 0.68 (0.66–0.69) |
| | <50 years | 2.8 (2.0–3.8) | 99.3 (99.2–99.4) | 24.8 (18.4–32.3) | 92.8 (92.5–93.2) | 0.62 (0.58–0.67) |
| | ≥50 years | 20.4 (18.5–22.4) | 97.2 (97.0–97.4) | 29.7 (27.0–32.4) | 95.5 (95.2–95.7) | 0.70 (0.69–0.72) |
| All cancers | All ages | 21.2 (19.8–22.6) | 97.8 (97.7–97.9) | 41.4 (39.1–43.7) | 94.4 (94.2–94.6) | 0.74 (0.73–0.75) |
| | <50 years | 6.1 (4.9–7.4) | 99.2 (99.0–99.3) | 37.3 (31.2–43.8) | 92.8 (92.5–93.2) | 0.70 (0.67–0.74) |
| | ≥50 years | 32.5 (30.4–34.6) | 96.9 (96.7–97.1) | 42.0 (39.5–44.5) | 95.5 (95.2–95.7) | 0.76 (0.75–0.78) |

PPV, NPV, sensitivity and specificity are calculated for a cutoff of ≥35 U/ml. Accuracy characteristics for "non-ovarian" cancer were calculated following exclusion of patients with ovarian cancer.

AUC, area under the curve; CA125, cancer antigen 125; CI, confidence interval; NPV, negative predictive value; PPV, positive predictive value.

**Table 4. Cancers diagnosed in women without ovarian cancer.**

| Cancer type (ICD10 codes) | N | N < 35 U/ml (%) | N ≥ 35 U/ml (%) |
|---|---|---|---|
| Unknown primary (C80) | 46 | 8 (17) | 38 (83) |
| Secondary: respiratory and digestive (C78) | 23 | 9 (39) | 14 (61) |
| Pancreas (C25) | 93 | 47 (51) | 46 (49) |
| Lung (C34) | 104 | 55 (53) | 49 (47) |
| Liver, biliary (C22, C23, C24) | 34 | 21 (62) | 13 (38) |
| Uterus (C54, C55, D39.0) | 132 | 84 (64) | 48 (36) |
| Upper GI (C15, C16, C17, D37.1, D37.2) | 66 | 46 (70) | 20 (30) |
| Lower GI (C18, C19, C20, C21,D37.3,D37.4,D37.5) | 255 | 197 (77) | 58 (23) |
| Hematological (C81, C82, C83, C84, C85, C90, C91, C92, C96, D45, D46, D47) | 112 | 83 (74) | 29 (26) |
| Kidneys, urinary tract (C64, C65, C66, C67, D41) | 78 | 65 (83) | 13 (17) |
| Breast (C50) | 154 | 142 (92) | 12 (8) |
| Other | 224 | 180 (80) | 44 (20) |
| **Total** | **1,321** | **937 (71)** | **384 (29)** |

"Other" consists of cancers with fewer than 10 cases with CA125 values ≥35 U/ml. The cancers included in this group and their frequencies are shown in S4 Table.

CA125, cancer antigen 125; ICD-10, International Classification of Diseases, 10th revision.

2.0%, whereas the incidence in women with a CA125 ≥35 U/ml, which equates to the PPV for non-ovarian cancers, was 12.3% (95% CI 11.2–13.5) (Table 3). This varied markedly between the <50 years group (PPV 2.8%, 95% CI 2.0–3.8) and ≥50 years group (PPV 20.4%, 95% CI 18.5–22.4). The PPV for all cancers was 21.2% (95% CI 19.8–22.6). Almost half of patients diagnosed with pancreatic and lung cancer in our cohort had CA125 levels ≥35 U/ml (Table 4).

## The probability of cancer by CA125 level

Fig 2 shows the relationship between CA125 level and the estimated probability of cancer, derived from logistic regression analyses. A CA125 level of 53 U/ml equated to a probability of 3% (95% CI 2.6–3.5) for ovarian cancer, whereas a CA125 level of 18 U/ml equated to a probability of 3% (95% CI 2.8–3.2) for all cancer. In a subanalysis in which invasive ovarian cancer formed the outcome, a CA125 level of 68 U/ml equated to a 3% probability.

Repeating the analysis in the <50 years and ≥50 years groups revealed that a much higher CA125 level was required to reach the 3% probability for ovarian cancer in the <50 years group (89 U/ml) than ≥50 years group (39 U/ml) (S4 Fig).

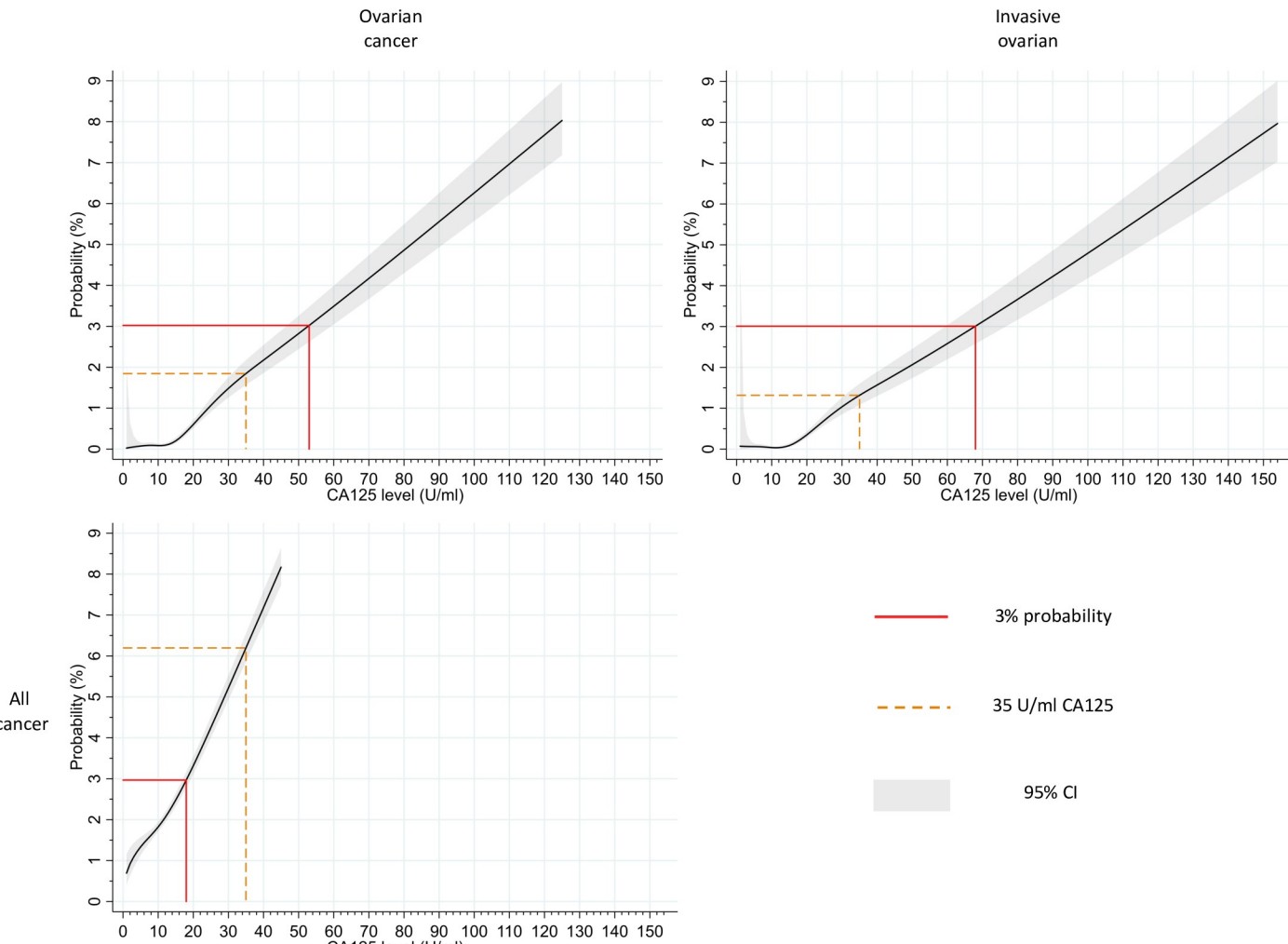

**Fig 2. Relationship between CA125 level and estimated probability of ovarian cancer, invasive ovarian cancer, and all cancers.** Estimated probabilities up to 8% are shown for ovarian cancer, invasive ovarian cancer, and all cancers. CA125 levels that correspond to the closest integer probabilities of 3% are indicated. The probabilities, which equate to a CA125 level of 35 U/ml, are also marked. Confidence intervals (95%) are displayed. Graphs showing probabilities at an extended range of CA125 values (up to 500 U/ml) for ovarian cancer, invasive ovarian cancer, and all cancer are included in S1 Fig, S2 Fig, and S3 Fig, respectively. Data used to construct these graphs (up to a CA125 level of 1,000 U/ml) are available via the University of Cambridge Repository [27]. CA125, cancer antigen 125; CI, confidence interval.

## The probability of ovarian cancer by age and CA125 level

**Fig 3** illustrates the relationship between CA125 level and the estimated probability of ovarian cancer at specific ages, derived from a logistic regression analysis. The probability of ovarian cancer at a given CA125 level varied markedly by age. The CA125 level required to reach the 3% ovarian cancer probability threshold fell from 104 U/ml in 40-year-old women to 32 U/ml in 70-year-old women. Similar age trends were noted when the analysis was repeated for invasive ovarian cancer (S5 Fig) and all cancer (S6 Fig).

## Discussion

In this cohort study of over 50,000 women who underwent CA125 testing in English general practice, 10.1% of those with a CA125 at or above the conventional cutoff (35 U/ml) were

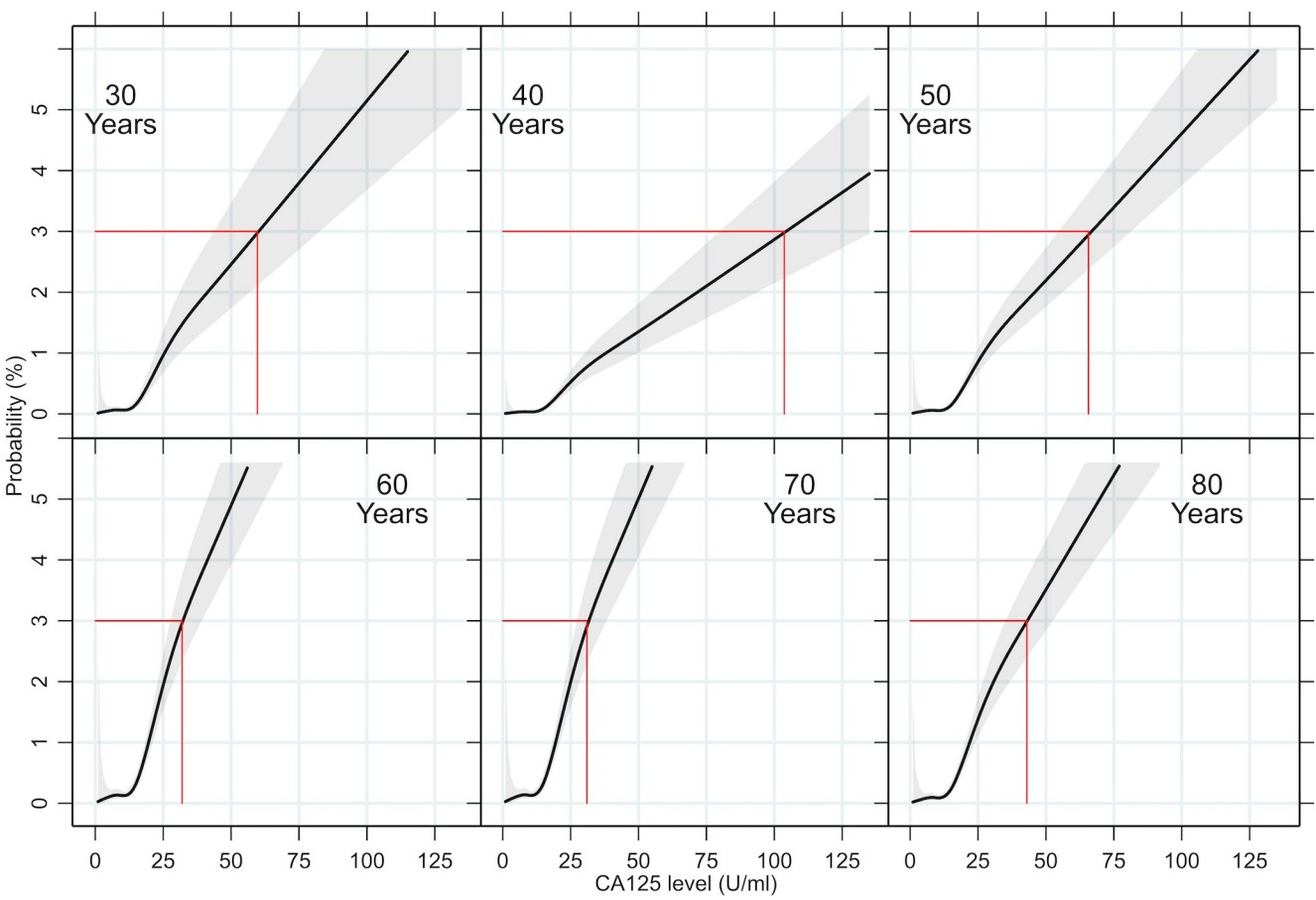

**Fig 3. Relationship between CA125 level and estimated probability of ovarian cancer for women of different ages.** Estimated ovarian cancer probabilities are shown in relation to CA125 level for women of 30, 40, 50, 60, 70, and 80 years of age. CA125 levels that correspond to the closest integer probabilities of 3% are indicated in red. Confidence intervals (95%) are displayed. Data used to construct these graphs (up to a CA125 level of 1,000 U/ml) are available via the University of Cambridge Repository [27]. CA125, cancer antigen 125.

diagnosed with ovarian cancer, and 12.3% were diagnosed with a different cancer. Almost a third of women aged ≥50 years with a CA125 ≥35 U/ml were diagnosed with some form of cancer. A CA125 level of 53 U/ml equated to an overall ovarian cancer probability of 3%—the threshold at which the UK NICE advocates urgent investigation or referral in symptomatic women. Marked variation was noted between women of different ages, with the 3% probability reached at lower CA125 levels in 70-year-old women than younger or older women.

## Study limitations

This study relied on coded routinely collected data, so it was not possible to determine exactly why CA125 tests were requested. However, the only indication for CA125 testing in UK primary care is to investigate symptoms of possible ovarian cancer. We identified symptoms recorded before CA125 testing, which may have been the trigger for testing. In contrast to CA125 results, which are automatically transferred into the GP system from laboratories, symptoms are not always coded but instead are often recorded in the free text within the GP record, which cannot be accessed for research purposes [28]. We did not restrict our analysis to women with a coded ovarian cancer symptom as this could introduce bias, given that symptoms are more likely to be coded if they are severe or persistent [28].

Our results reflect real-world use of CA125 in English general practice. How CA125 is used in primary care in other countries may differ from practice in England. Baseline CA125 levels may also be affected by population characteristics, such as ethnicity, smoking status, and past medical history [29]. We did not include these variables in our analysis as our aim was to develop simple models that allow the estimated probability of cancer to be reported alongside the CA125 result in general practice, without the need to collect further detailed information from the patient. Although this study has significant international relevance, caution is needed when translating our findings to other countries and healthcare systems.

We report ovarian cancer stage at diagnosis, but CA125 diagnostic accuracy was not analyzed by stage. As CA125 tests were performed at variable intervals in the 12 months preceding diagnosis, such an analysis is likely to be misleading as an unknown number of cancers will have progressed during that period.

We have employed restricted cubic splines to model the nonlinear nature of the relationship between both age and CA125 level and cancer diagnoses. Although these provide a flexible approach to parameterizing the fitted relationships, there is a large degree of uncertainty at the extremes of age and CA125 level, and so the cancer probabilities for very old and young women and those with very low CA125 levels should be treated with caution, and the large CI noted.

## Results in the context of other studies

In their 2011 ovarian cancer guidelines, NICE estimated that 0.81% of symptomatic primary care women with a CA125 $\geq$35 U/ml would have ovarian cancer [8]. Economic modeling and the recommendation for sequential testing with CA125 followed by ultrasound if the CA125 were abnormal was predicated on this estimate. Our findings indicate that the PPV is more than 12 times higher than estimated. This is consistent with the only other UK report of the PPV of CA125 in primary care, which found that 16 out of 152 women (11%) with a raised CA125 level had ovarian cancer [13]. The sensitivity of CA125 for ovarian cancer in our study was slightly lower—and the specificity higher—than reported in studies in which testing was performed in women with a pelvic mass prior to surgery in secondary care [30]. This is to be expected as tests generally have lower sensitivity and higher specificity in populations with a lower disease prevalence—the spectrum effect [10]. As anticipated, the PPVs for ovarian cancer in our cohort were lower than in secondary care patients with pelvic masses [31] and higher than in asymptomatic screening populations [32].

One of the most striking findings in our study was the high incidence of non-ovarian cancers in those with elevated CA125 levels, particularly in women aged 50 years or older. This reflects the nonspecific nature of ovarian cancer symptoms and also that CA125 is frequently raised in women with a variety of non-ovarian malignancies [12]. Crawford and colleagues reported that 16 out of 152 women (11%) referred from primary care with a raised CA125 were diagnosed with a non-ovarian cancer [13]. Furthermore, in asymptomatic screening populations, a higher incidence of non-ovarian cancers has been noted in women with raised CA125 levels (6.9%) than with normal CA125 levels (1.6%) [33].

We found that the estimated probability of ovarian cancer for a given CA125 level rose with age to peak in women in their seventies, which mirrors UK age-specific cancer incidence rates [34]. The exception was very young women—the probability of ovarian cancer at a given CA125 level was higher in women aged 30 than aged 40. This probably reflects GP testing practices in very young women (in whom ovarian cancer is extremely rare), with GPs having a strong reason to request a CA125 test in these women, thereby raising the pretest probability.

## Clinical interpretation of the findings

Of the CA125 tests performed, 39% were in women <50 years of age; however, ovarian cancer is predominantly a disease of older and postmenopausal women. This is reflected in our findings, as only 18% of ovarian cancers and 11% of the invasive subtype occurred in women under 50. All measures of test performance, save for the NPV, were worse in women under 50 years than 50 years and over, even when borderline malignancies (which were more common in the younger age group) were excluded. A greater proportion of invasive tumors in the <50 years group were mucinous epithelial and nonepithelial cancers, both of which have less propensity to elevate serum CA125 than other ovarian cancer types, likely contributing to poorer test performance in the younger age group [31]. The results of our regression analysis indicate that, overall, only 1 in 110 women <50 years with a CA125 of exactly 35 U/ml will have an ovarian cancer, and only 1 in 308 will have an invasive subtype. Investigating younger women for ovarian cancer when there is high suspicion is important, but given the low incidence of ovarian cancer and relatively poor test performance in women under 50 years, CA125 tests should be performed and interpreted with caution in this group.

The total number of non-ovarian cancers diagnosed in women with raised CA125 levels exceeded that of ovarian cancers, but the numbers of women with each type of non-ovarian cancer was small. In isolation, CA125 is unlikely to be a useful test for the detection of individual types of non-ovarian cancer in primary care, most of which have superior triage tests. However, given our study findings, a high CA125 level in a woman ≥50 years should raise a suspicion of non-ovarian cancer. Clinicians should consider these cancers and whether further investigation is required, particularly if ovarian cancer has been excluded. Research is needed to determine the most appropriate follow-up and testing strategy for these women in order to ensure prompt diagnosis.

When assessing test performance, it is standard practice to evaluate test characteristics using a cutoff, above which the test is deemed abnormal and below which it is deemed normal. As per convention, we have presented this for CA125, applying the standard ≥35 U/ml cutoff. However, where the probability of having a disease varies markedly with the test level, PPV is of limited value in informing decisions about individual patients, as it effectively provides an average probability of disease for all women with "abnormal" results. In this study, women with very high CA125 values had a very high probability of being diagnosed with cancer. Conversely, those with CA125 levels around the 35 U/ml cutoff had a much lower probability of being diagnosed with cancer than the PPV would appear to indicate. In this study, we have quantified the risk of cancer in individuals with specific CA125 values at specific ages. This should be of much more use clinically than PPVs.

Estimated cancer probabilities will allow women and clinicians to interpret their individual CA125 result and could inform health policy both in the UK and internationally. For example, NICE currently recommend that women with a CA125 ≥35 U/ml, whether 35 U/ml or 1,000 U/ml, should be referred for an ultrasound scan, whereas no further investigations for ovarian cancer are advocated in women with levels below the cutoff. Instead, our results could be used to triage women of different ages, selecting those with a high probability of ovarian cancer for expedited referral and investigation. Women with a probability in excess of the NICE risk threshold of 3% could be referred via the urgent cancer pathway for specialist gynecology assessment and/or imaging. Women with lower probabilities might, after discussion between clinician and patient, be investigated using routine ultrasound, recognizing the fact that patients would opt for cancer testing at risk levels as low as 1% [35]. As only a woman's age and CA125 level are required to determine the cancer probability from our results, this information could readily be incorporated into laboratory information technology (IT) systems,

reported alongside the CA125 level, and communicated to patients in clear terms, e.g., "1 in 30 women of your age who have the same CA125 level in general practice will have ovarian cancer."

Although we have focused on the UK NICE 3% probability threshold for urgent cancer referral, our results would also allow alternative thresholds for referral to be implemented. A lower probability threshold may lead to the detection of more cancers, but this would come at the cost of larger numbers of cancer-free women being referred and further investigated, which can have negative consequences such as increased patient anxiety and financial cost [36]. Conversely, employing a higher probability threshold would lead to fewer cancer-free women being referred unnecessarily, but more cancers may be missed. A full health economic evaluation would greatly improve understanding of the implications of applying different referral thresholds.

The Refining Ovarian Cancer Test Accuracy Scores (ROCkeTS) study, a large ongoing prospective study in the UK evaluating a range of diagnostic tests and algorithms for ovarian cancer in secondary care, may provide insight into the most appropriate post-CA125 testing strategy [37]. Any such strategy should take account of the high incidence of non-ovarian cancers in women with high CA125 levels, as pelvic ultrasound alone will miss many of these malignancies. Other imaging modalities such as computed tomography (CT), which can detect multiple CA125 elevating cancers including ovarian, lung, and pancreatic cancer [38–40] and which is already used in several countries to investigate symptomatic women with elevated CA125 levels [7], could be appropriate. Further research is also needed to determine whether CA125 re-testing in primary care should be performed in women who have a normal ultrasound scan but persistent symptoms, as there is evidence from screening studies that a rising CA125 is associated with a higher risk of ovarian cancer, even if ultrasound is normal [41].

## Conclusions

CA125 is a useful test for detecting ovarian cancer in primary care, particularly in women aged 50 years and over. Given the high incidence of non-ovarian cancers in women with elevated CA125 levels, clinicians should consider alternative cancers particularly when ovarian cancer has been excluded. The results of this study will enable patients and clinicians to interpret their CA125 result in terms of the probability of cancer at the pertinent CA125 level and age. The findings will also allow policy makers to provide recommendations for post-CA125 investigations on the basis of the probability of undiagnosed cancer, which could enable the expedited investigation and referral of those women most likely to have a cancer.

## Supporting information

**S1 Text. ISAC protocol.** ISAC, Independent Scientific Advisory Committee.
(PDF)

**S2 Text. Minor amendment to ISAC protocol (Dated: 02/07/2019).** ISAC, Independent Scientific Advisory Committee.
(PDF)

**S3 Text. Completed STARD checklist.** STARD, Standards for Reporting of Diagnostic Accuracy Studies.
(PDF)

**S4 Text. Logistic regression model specifications.**
(PDF)

**S1 Table. Read codes and terms used to identify CA125-tested women.**
(PDF)

**S2 Table. Ovarian cancer by stage of diagnosis.**
(PDF)

**S3 Table. Behavior and histology of ovarian tumors by age group (<50 years and ≥50 years).**
(PDF)

**S4 Table. Frequencies of non-ovarian cancers included in the "other" group in Table 4.**
(PDF)

**S1 Fig. Estimated probabilities of ovarian cancer at an extended range of CA125 levels.**
(PDF)

**S2 Fig. Estimated probabilities of invasive ovarian cancer at an extended range of CA125 levels.**
(PDF)

**S3 Fig. Estimated probabilities of all cancer at an extended range of CA125 levels.**
(PDF)

**S4 Fig. Relationship between CA125 level and estimated probability of ovarian cancer, invasive ovarian cancer and all cancers in women <50 years and ≥50 years of age.** Estimated probabilities up to 8% for each cancer type in women <50 and ≥50 years of age are shown, save for invasive cancer in the <50 years group, in which a CA125 of 317 U/ml was required to reach an 8% probability.
(PDF)

**S5 Fig. Relationship between CA125 level and estimated probability of invasive ovarian cancer for women of different ages.** Probabilities are shown in relation to CA125 level for women of 30, 40, 50, 60, 70, and 80 years of age. CA125 levels that correspond to the closest integer probabilities of 3% are indicated in red (not displayed in 40 years of age- reached at 191 U/ml). The 95% confidence intervals are displayed.
(PDF)

**S6 Fig. Relationship between CA125 level and estimated probability of all cancer for women of different ages.** Probabilities are shown in relation to CA125 level for women of 30, 40, 50, 60, 70, and 80 years of age. CA125 levels that correspond to the closest integer probabilities of 3% are indicated in red. The 95% confidence intervals are displayed.
(PDF)

## Acknowledgments

This work uses data provided by patients and collected by the NHS as part of their care and support. We are grateful to Mrs. Margaret Johnson for her input into study development and to Professor Greg Rubin for his helpful comments during paper preparation.

The views expressed are those of the authors and not necessarily those of the National Institute for Health Research (NIHR), the Department of Health and Social Care, or Cancer Research UK.

## Author Contributions

**Conceptualization:** Garth Funston, Willie Hamilton, Emma J. Crosbie, Fiona M. Walter.

**Data curation:** Garth Funston, Brian Rous.

**Formal analysis:** Garth Funston, Gary Abel.

**Funding acquisition:** Garth Funston, Willie Hamilton, Fiona M. Walter.

**Investigation:** Garth Funston.

**Methodology:** Garth Funston, Willie Hamilton, Gary Abel, Brian Rous, Fiona M. Walter.

**Project administration:** Garth Funston.

**Supervision:** Fiona M. Walter.

**Writing – original draft:** Garth Funston.

**Writing – review & editing:** Garth Funston, Willie Hamilton, Gary Abel, Emma J. Crosbie, Brian Rous, Fiona M. Walter.

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
