## [Editor Report · Decision Letter 0]

7 Feb 2020

Dear Dr Funston, 

Thank you for submitting your manuscript entitled "The diagnostic performance of CA125 for the detection of ovarian and non-ovarian cancer in primary care: a population-based cohort study" for consideration by PLOS Medicine.

Your manuscript has now been evaluated by the PLOS Medicine editorial staff [as well as by an academic editor with relevant expertise] and I am writing to let you know that we would like to send your submission out for external peer review.

Kind regards,

Adya Misra, PhD,

Senior Editor

PLOS Medicine

---

## [Decision Letter · Decision Letter 1]

17 May 2020

Dear Dr. Funston,

Thank you very much for submitting your manuscript "The diagnostic performance of CA125 for the detection of ovarian and non-ovarian cancer in primary care: a population-based cohort study" (PMEDICINE-D-20-00302R1) for consideration at PLOS Medicine. 

[LINK]

In light of these reviews, I am afraid that we will not be able to accept the manuscript for publication in the journal in its current form, but we would like to consider a revised version that addresses the reviewers' and editors' comments. Obviously we cannot make any decision about publication until we have seen the revised manuscript and your response, and we plan to seek re-review by one or more of the reviewers. 

We expect to receive your revised manuscript by Jun 05 2020 11:59PM. Please email us (plosmedicine@plos.org) if you have any questions or concerns.

We look forward to receiving your revised manuscript. 

Sincerely,

Emma Veitch, PhD

PLOS Medicine

On behalf of Clare Stone, PhD, Acting Chief Editor,

PLOS Medicine

plosmedicine.org

*Some minor reformatting is needed to the article structure to fit better with PLOS Medicine's format. In the abstract, we'd ask the authors to restructure this using the PLOS Medicine headings (Background, Methods and Findings, Conclusions) - "Methods and Findings" is a single subsection.

*In the last sentence of the Abstract Methods and Findings section, please describe some of the key limitation(s) of the study's methodology.

*At this stage, we ask that you include a short, non-technical Author Summary of your research to make findings accessible to a wide audience that includes both scientists and non-scientists. The Author Summary should immediately follow the Abstract in your revised manuscript. This text is subject to editorial change and should be distinct from the scientific abstract. Please see our author guidelines for more information: https://journals.plos.org/plosmedicine/s/revising-your-manuscript#loc-author-summary

*Ideally, please reformat the in-text reference callouts to use the PLOS Medicine style (numerals in square brackets) rather than superscript numerals- this should be fairly quick and easy if using referencing software.

*Did your study have a prospective protocol or analysis plan? Please state this (either way) early in the Methods section.

*The two subsections at the very end of the paper, 'What is already known on this topic' and 'What this study adds' should be deleted (and any unique material from there incorporated into the paper) as these aren't needed for PLOS Medicine publication.

*It's good that the paper has been reported according to both STARD and RECORD reporting guidelines - we'd suggest the authors include as supporting information the completed checklist for just one of these, ideally STARD as that would be the relevant guideline for the overall framework of the study. 

Comments from the reviewers:

Reviewer #1: I confine my remarks to statistical aspects of this paper.

Overall, I think they were outstanding and I wish other authors of similar studies could be made to read these methods and adapt them. I've been saying stuff like this for years but little has changed. Good to see.

But I do have some issues to resolve before I can recommend publication.

NOTE: There were neither page numbers nor line numbers - this makes things tricky for the reviewer

Intro: Give numbers for stage II and III

Model building: The sentence starting "The number of polynomial terms ...." Whether to use splines or polynomials is an interesting question. Using AIC to decide is one reasonable way. I tend to like splines, but polynomial terms can also work. But I would set an upper limit on the order of they polynomial (maybe cubic but no higher) because beyond that they lose the main advantage over splines - interpretability. Please list the terms used here (and give details in the appendix).

 Rather than dichotomize age, leave it contnuous and investigate it the same way you did CA125. Then include the terms (spline or polynomial) in the model. This would give a probability for every combination of age and CA. And the plts could be changed to trellis plots. 

Section on cutoff: Maybe add a discussion of why 3% was chosen and the costs of the diferent errors.

figure 2 is too small to read, at least on a printout. I needed a magnifying glass. 

Peter Flom

Reviewer #2: This is a very informative study reporting the estimated probabilities of ovarian cancer and all cancers are reported for a wide range of CA125 levels. The findings will allow assessment of the probability of undiagnosed cancer, other than ovarian, enabling the expedited investigation and referral in the UK. 

The only critique is that the first paragraph of Abstract is very confusing and should be revised.

Reviewer #3: Funston G and colleagues present their findings on the diagnostic performance of Cancer Antigen 125 (CA125) to detect ovarian and non-ovarian cancer. Being able to effectively diagnosis ovarian cancer at early stages has remained one of the most significant challenges in managing ovarian cancer. Nearly 80% of patients are diagnosed at late stages, which confers a poor prognosis. The authors recognize that most cancer care starts with the general practitioners and there is a urgent need to better define cancer risk associated with elevated CA125 levels. Specifically, while the risk threshold for investigating a potential cancer was been redefined at 3%, the guidance for ovarian cancer remains largely dependent on CA125 levels (35 U/mL). The authors performed a large (n=50,780) retrospective study on patients being treated in the English system. The data is intriguing and will provide strong evidence to shift clinical interpretation of CA125 levels. The manuscript is well written, concise, and clear. Overall, the research study significantly contributes to the field of ovarian cancer. There are a couple of noted weaknesses that should be addressed. 

1) The authors note that a limitation of this study was that only contains patients being evaluated in the English system. This limitation should be expanded upon at minimum in the discussion and potentially in the results. For instance, there several confounding factors that contributes to CA125 levels including race, smoking, hysterectomy, menstrual cycle, etc. (PMID: 11352859). Changes in menstrual cycle can account for CA125 change up to 22%. It is unclear from the manuscript whether beyond age these factors contributed or detracted from the findings.

2) Different histological subtypes for ovarian cancer should be noted, because different histotypes are not likely to have the same CA125 threshold. It is interesting that in women <50 y.o. required a CA-125 level of 179 U/mL to equate to a 3% probability of invasive ovarian cancer, which is likely due to different (other than high grade serous) and less invasive histotypes being more prevalent in pre-menopausal women.

3) Figure 2 is difficult to read, the text is too small, and the y-axis different for each graph.

[LINK]

---

## [Decision Letter · Decision Letter 2]

11 Aug 2020

Dear Dr. Funston,

Thank you very much for re-submitting your manuscript "The diagnostic performance of CA125 for the detection of ovarian and non-ovarian cancer in primary care: a population-based cohort study" (PMEDICINE-D-20-00302R2) for review by PLOS Medicine.

I have discussed the paper with my colleagues and the academic editor and it was also seen again by the statistical reviewer. I am pleased to say that provided the remaining editorial and production issues are dealt with we are planning to accept the paper for publication in the journal.

[LINK]

We look forward to receiving the revised manuscript by Aug 18 2020 11:59PM. 

Sincerely,

Clare Stone, PhD

Acting Chief Editor

PLOS Medicine

plosmedicine.org

Requests from Editors:

Please avoid use of itals for emphasis

The checklist should have sections and paragraphs instead of page numbers as these can change during formatting etc. 

Comments from Reviewers:

Reviewer #1: The authors have addressed my concerns and I now recommend publication.

Peter Flom

[LINK]

---

## [Editor Report · Decision Letter 3]

11 Sep 2020

Dear Dr Funston, 

On behalf of my colleagues and the academic editor, Dr. Steven D Shapiro, I am delighted to inform you that your manuscript entitled "The diagnostic performance of CA125 for the detection of ovarian and non-ovarian cancer in primary care: a population-based cohort study" (PMEDICINE-D-20-00302R3) has been accepted for publication in PLOS Medicine. 

PRODUCTION PROCESS

PRESS

PROFILE INFORMATION

Thank you again for submitting the manuscript to PLOS Medicine. We look forward to publishing it. 

Best wishes, 

Clare Stone, PhD

Managing Editor

PLOS Medicine

plosmedicine.org